# Corrosion Activity of Carbon Steel B450C and Low Chromium Ferritic Stainless Steel 430 in Chloride-Containing Cement Extract Solution

Lucien Veleva [1],* , David Bonfil [1], Ángel Bacelis [1], Sebastian Feliu, Jr. [2], Marina Cabrini [3] and Sergio Lorenzi [3]

[1] Applied Physics Department, Center for Investigation and Advanced Study (CINVESTAV), Merida 97310, Mexico; david.bonfil@cinvestav.mx (D.B.); angel.bacelis@cinvestav.mx (Á.B.)
[2] Surface Engineering Corrosion and Durability Department, National Center for Metallurgical Research (CENIM-CSIC), 28040 Madrid, Spain; sfeliu@cenim.csic.es
[3] Department of Engineering and Applied Sciences, INSTM RU Bergamo and University of Bergamo, Viale Marconi 5, 24044 Dalmine, Italy; marina.cabrini@unibg.it (M.C.); sergio.lorenzi@unibg.it (S.L.)
* Correspondence: veleva@cinvestav.mx; Tel.: +52-999-9429400

**Abstract:** The carbon steel B450C and low chromium SS 430 ferritic samples were exposed for 30 days to chloride-containing (5 g $L^{-1}$ NaCL) cement extract solution. The initial pH $\approx$ 13.88 decreased to pH $\approx$ 9.6, associated mainly with the consumption of $OH^-$ ions and the formation of $\gamma$-FeOOH, $\alpha$-FeOOH, $Fe_3O_4$ and $Cr(OH)_3$, as suggested by XRD and XPS analysis, in the presence of $CaCO_3$ and NaCl crystals. The deep corrosion damages on B450C were observed around particles of Cu and S as local cathodes, while the first pitting events on the SS 430 surface appeared after 30 days of exposure. The change in the activity of each type of steel was provided by the potentiodynamic polarization curves (PDP). Two equivalent electrical circuits (EC) were proposed for quantitative analysis of EIS (Nyquist and Bode diagrams). The calculated polarization resistance ($R_p$), as an indicator of the stability of passive films, indicated that SS 430 presented relatively constant values, being two-three orders of magnitude higher than those of the carbon steel B450C. The calculated thickness (*d*) of the SS 430 passive layers was $\approx$0.5 nm and, in contrast, that of the B450C passive layers tends to disappear after 30 days.

**Keywords:** carbon steel; stainless steel; concrete pore; cement extract; chlorides; corrosion tests; XPS; EIS

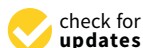



## 1. Introduction

Carbon and stainless steel are being used to reinforce concrete and because of the steel-concrete union may improve the strength of concrete against a variety of mechanical forces. The corrosion stability of these steels is acquired during the cement hydration process, when the Portland cement provides a high alkaline pH $\approx$ 13 (because of the formation of calcium hydroxide $Ca(OH)_2$, portlandite) during the curing time, and the passive layer is formed on the steel surface [1–3]. As a consequence, the reinforced concrete supports non-aggressive environments for a long service life, and the steel does not present corrosion, when the pH of the concrete is always relatively stable. On the other hand, because of the self-generation properties of stainless steel passivation (mainly due to the ability of the chromium rich oxide layer) [4,5], even when the alkalinity of concrete diminishes and the aggressive chloride ions ($Cl^{-1}$) have reached the steel surface (for example, in a marine environment), stainless steel concrete reinforcement is preferred to the traditional carbon (mild) steel [6–8]. In the presence of Cl-ions, the passivated (mild) steel may suffer deterioration, since layers of oxide/hydroxides appear on the iron (Fe) surface [1], even in the highly alkaline concrete environment (pH 12–13); this fact is followed by the corrosion initiation and then its propagation [8–11]. Thus, the durability (service life) of the reinforced concrete will diminish, due to the thickening of the iron corrosion product layer formed on

it, and the detachment of the concrete cover will occur as a consequence of the mechanical internal stresses. A specific concentration of chlorides is needed for the initiation of the chloride-induced corrosion, known as the chloride threshold value. For stainless steels, the reported threshold value ($Cl^-/OH^- > 1$) is higher than that needed for carbon steel rebars ($Cl^-/OH^- < 1$), during their exposure to alkaline solutions [12,13]. Once corrosion is initiated, the expansion of the voluminous corrosion products exerts tensile stresses within the cementitious matrix, causing cracking and even spalling [14,15].

The passive-layer breakdown is a very complex phenomenon that is not fully understood. A review of the models and the influence of halogen anions is given by Szklarska-Smialowska [16]. Several events are considered, such as: thinning of the passive film; "mechanical" or electrocapilllary breakdown of the film, as a precursor of pitting corrosion initiation; penetration of Cl-ions trough the oxide passive layer; and dissolution of the metal (Fe) matrix. However, there is criticism of these models because experimental results do not conform to fit to them [17].

With regard to instrumental techniques and data interpretation [18,19], corrosion of the steel embedded in concrete has been the subject of extensive research and discussion because of the difficulties in experimental measurements, including: electrode cell design and position of the electrodes; large potential drop (IR) in the concrete; restriction of oxygen diffusion; and development of macro- and micro-corrosion cells, etc. To avoid the mentioned difficulties, as an alternative approach, a variety of alkaline solutions that model the steel-concrete-pore environment have been proposed [20–33], although some models differ in the results obtained. The model solutions allow one to obtain comparative results and the control of some experimental parameters, which is difficult to accomplish in reinforced concrete samples.

The most common model solution is saturated $Ca(OH)_2$, however, the composition of concrete pore solution is complex [29–31], since it includes a number of other compounds besides $Ca(OH)_2$. The passive layers on AISI 316, grown during the exposure to saturated $Ca(OH)_2$ and cement extract (CE) solutions, revealed that each model solution simulates the concrete-pore environment in a different way [29,30]; a more resistant layer, homogeneous in thickness and porosity, was formed in CE solution, which is due to its ionic, distinctive composition compared to that of saturated $Ca(OH)_2$. This fact was confirmed by AFM (atomic force microscopy) and XPS, as well by cyclic voltammetry.

This research compares the corrosion behavior of two commercial steels, the Italian carbon steel B450C and the low chromium ferritic Finnish stainless steel SS 430, exposed for 30 days to chloride-containing (5 g L$^{-1}$ NaCl) cement extract unbuffered solution, in order to simulate the environment at the steel-concrete-pore interface. In our previous study, the corrosion activity of these steels was performed in free-chloride cement extract solution [34]. It was very important to establish the corrosion development of each steel in the traditional pH of the concrete-pore environment, since both steels have been proposed as reinforcement in a lower pH of concrete than that of the traditional pH. Different techniques and methods have been applied to help in this aspect. Potentiodynamic polarization curves (PDP) and two non-destructive electrochemical techniques were performed, viz. free corrosion potential monitoring at open circuit potential (OCP) and electrochemical impedance spectroscopy (EIS). The surfaces of the steels were characterized by scanning electron microscopy (SEM), X-ray photoelectron spectroscopy (XPS) and X-ray diffraction (XRD) techniques. To our knowledge, no other research on this topic has been previously undertaken.

## 2. Materials and Methods

### 2.1. Samples and Solution Preparation

Carbon steel construction material (B450C) was supplied by Pittini Group (Gemona del Friuli, Italy) and commercial low chromium ferritic stainless steel (SS 430) was provided by Outokumpu (Espoo, Finland). Before the immersion tests, flat samples were cut (2 cm × 1 cm × 0.1 cm), abraded with wet SiC to 4000 grit (using ethanol as a lubricant to prevent oxidation), then sonicated for 10 min (Branson 1510, Branson Ultrasonics Co., Danbury, CT,

USA) and dried in air (at room temperature). The elemental composition of the steels is present in Table 1.

**Table 1.** Compositions (wt.%) of SS 430 ferritic steel and B450C carbon steel, according to suppliers Outokumpu (Finland) and Pittini Group (Italy).

| Element (wt.%) | C | Cr | N | Cu | P | S | Fe |
|---|---|---|---|---|---|---|---|
| SS 430 | 0.05 | 16.2 | - | - | - | - | Balance |
| B450C | 0.22 | - | 0.12 | 0.8 | 0.5 | 0.5 | Balance |

The surfaces of the control samples were characterized by Scanning Electron Microscopy (SEM-EDS, XL–30 ESEM-JEOL JSM-7600F, JEOL Ltd., Tokyo, Japan). Figure 1 shows the SEM images of ferritic SS 430 (Figure 1a) and carbon steel B450C (Figure 1b) surfaces of control samples, as well as the EDS elemental analysis (Table 2).

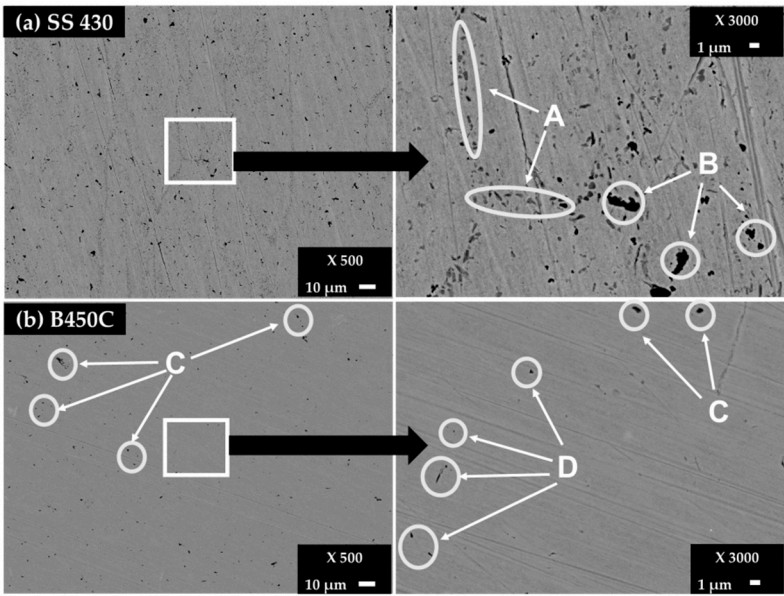

**Figure 1.** SEM images (×500) of surface steel control samples and zoom (×3000) of areas of interest: (**a**) SS 430 and (**b**) carbon steel B450C.

**Table 2.** EDS surface analysis (wt.%) of SS 430 (Figure 1a) and carbon steel B450C (Figure 1b) control samples.

| Element | Element | C | Cr | Mn | Si | O | V | Cu | N | S | Fe |
|---|---|---|---|---|---|---|---|---|---|---|---|
| | General | 1.83 | 16.32 | 0.65 | 0.6 | 0.34 | 0.28 | - | - | - | 79.97 |
| SS 430 | A | **3.04** | **24.03** | - | 0.25 | 0.62 | **0.7** | - | **3.27** | - | 68.09 |
| | B | **17.19** | **9.21** | - | **28.86** | 1.35 | - | - | - | - | 43.39 |
| | General | 2.36 | - | 0.81 | - | 0.47 | - | - | - | - | 96.36 |
| B450C | C | **5.02** | - | **1.31** | 0.41 | 1.44 | - | **0.83** | - | **0.38** | 90.36 |
| | D | **8.49** | - | 0.73 | **6.46** | 0.46 | - | - | - | - | 83.85 |

Besides the Cr-content (≈16 wt.%) on the ferritic SS 430 surface, supplying its pitting corrosion resistance [35,36], two zones were identified on the SS 430 surface (Figure 1, Table 2). The multiples dots in the A zones revealed a higher Cr content (24.03%) in the presence of C (3.04%) and N (3.27%), probably of chromium nitride and carbide phases, precipitated at the grain boundaries because of the lower solubility of C and N as well as the fast diffusivity of Cr in the ferrite phase [37]. On the other hand, the existence of carbide phase $(Cr, Fe)_7C_3$ and Cr-nitride are also reported [36,37]. The aggregates in the B zones

reveal a high carbon content (17.19%) in the presence of Si (28.86%) that may be attributed to silicon carbide (SiC). The observed V (0.7%, zone B) probably has replaced the Cr sites in the lattice of Cr-C-N crystal structure, forming precipitates of the stable carbide phase ($V_6C_5$) and VN-nitride phase, which block and prevent the grains from growing larger as well increasing the ductility, hardness and strength of the ferritic steel [38].

Carbon steel B450C control samples present black dots in two zones, named C and D (Figure 1b). The C zones, according to their EDS analysis (Table 2), contain Carbon (5.02%), Mn (1.31%) and a lower content of S (0.38), which may be considered as a part of the phases of MnS and Mn3C reported for this type of steel [39,40]. The existence of Cu may be attributed to the quality of the scrap used to produce this carbon steel [40]. The content of Si (6.46%), in the presence of Carbon (8.49%) in the D zone, was attributed to the SiC phase (Figure 1b, Table 2). Manganese and silicon are always present in carbon steel, although not explicitly reported by the supplier.

The cement extract solution was prepared to simulate non-carbonated concrete-pore environment, from a 1:1 wt./wt. mixture of Portland cement type I, produced by CEMEX (CEMEX, S.A.B. de C.V., San Pedro Garza García, N.L., México), and ultrapure deionized water (18.2 MΩ·cm). The mixture was left for 24 h to hydrate in a closed container (to prevent carbonation by absorption of $CO_2$ from the air). Then, the supernatant was filtered to remove particles, ultimately with a 0.2-μc syringe filter (Whatman, Kent, UK). The chemical composition of the cement, according to the producer, and that of the extract solution after filtration, were reported previously [29,34,41]. The authors report that in Portland cement type I the highest content (wt.%) corresponds to CaO (66.84–58.4), followed by $SiO_2$ (21.35–22.30), $Al_2O_3 \approx 4.7$), $Fe_2O_3$ (2.89), $SO_3$ (2.42), MgO (1.16) and, at low contents, $K_2O$ (0.39–0.35) and $Na_2O$ (0.08–0.28). The extract cement solution contains as ions (mmol $L^{-1}$): $Ca^{2+}$ (6.4), $K^+$ (35.1), $Na^+$ (18.3) and $OH^-$ (56.4), as well as $SO_4^{2-}$. NaCl was added to cement extract solution in the amount of 5 g $L^{-1}$. The initial pH value of this model solution was 13.88, and it was regularly checked by a pH meter during the immersion of the steel samples.

### 2.2. Immersion Test and Surface Characterization

The triplicate steel samples (0.8 $cm^2$ working area) were immersed in 50 mL of model cement extract solution for a period up to 720 h (30 days), according to the guide for laboratory immersion corrosion testing of metals of ASTM G31-12a [42]. After 168 h (7 days) and 720 h (the end of the experiment), the samples were withdrawn, rinsed with deionized water and dried in air at room temperature (21 °C).

SEM-EDS images provided the damage on the steel surfaces after the removal of the corrosion layers, according to the cleaning solutions and procedures recommended by ASTM G1-03 standard [43]. The elemental composition of the corrosion product layer was provided by X-ray Photoelectron Spectrometer (XPS, K-Alpha, Thermo Scientific, Waltham, MA, USA), equipped with a monochromatic Al K-alpha radiation source (1486.6 eV): the pass energy was 50 eV and the energy step size was 0.1 eV for the scan of XPS spectra, while for survey spectra they were 100 eV and 1 eV, respectively. The spectra were calibrated by setting the main line for the O1s signal of oxygen in oxides at 530.2 eV, according to the procedures suggested by Yamashita and Hayes for transition metal oxides [44]. The XPS spectra were obtained after sputtering the specimens' surface with a scanning argon-ion gun for 15 s.

### 2.3. Electrochemical Measurements

A typical three-electrode cell configuration inside a Faraday cage and an Interface-1000E potentiostat/galvanostat/ZRA (Gamry Instruments, Philadelphia, PA, USA) were used for electrochemical experiments (at room temperature, 21 °C). Steel plates were the working electrodes (0.8 $cm^2$), while the Pt plate was used as an auxiliary and a saturated calomel electrode (SCE) was used as the reference electrode. Electrochemical Impedance Spectroscopy (EIS) measurements were performed at the open circuit potential (OCP),

applying an alternating current (AC) signal of $\pm 10$ mV amplitude in a frequency range from 100 kHz to 10 mHz, and with a sampling size of 10 data points/decade. Nyquist and Bode EIS diagrams were recorded at different immersion periods: 10 min (initial), 24, 168, 360, 504 and 720 h (30 days). The data were analyzed with Gamry Echem Analyst® (version 7.1, Philadelphia, PA, USA). Potentiodynamic polarization curves (PDP) were collected after achievement of a stable initial open circuit potential (OCP) of the electrodes immersed in the cement extract solution, by applying cathodic polarization to $-500$ mV and anodic to $+1500$ mV, with a scan rate of 1 mV s$^{-1}$. The obtained PDP curves were examined with the Gamry Echem Anayst® (version 7.1, Philadelphia, PA, USA).

## 3. Results and Discussion

### 3.1. Change in Time of pH of Chloride-Containing Cement Extract Solution during Exposure of Steels

Table 3 presents the change in time of pH of cement extract solution during the exposure for 720 h (30 days) of SS 430 and carbon steel B450C samples. The initial pH of 13.88 tended towards less alkaline values since the first 24 h, maintaining almost constant pH $\approx 9.6$ until the end of the test. The decrease in pH was associated with the OH$^-$ ions consumption, needed for the formation of hydroxides/oxyhydroxides steel corrosion products. Given 50 mL volume of the tested solution, the dissolution of $CO_2$ gas may promote the formation of carbonic acid ($H_2CO_3$) [45]. For $7 < pH < 10$, according to study [46], the $HCO_3^-$ ions are predominate and corrosive to iron; thus, the released $Fe^{2+}$ ions attract the OH$^-$ from the medium (a decrease in pH), mainly forming iron (II) hydroxide. A similar behavior of the cement extract solution pH was reported in the absence of chlorides [34]. According to the Pourbaix diagram at pH values lower than 11.7, the carbon steel lost its passive state [1].

**Table 3.** Change in time of pH of chloride-containing (5 g L$^{-1}$ NaCl) cement extract solution, during the immersion of SS 430 and carbon steel B450C for 720 h (30 days).

| pH vs. Time (h) | Initial | 24 | 168 | 360 | 504 | 720 |
|---|---|---|---|---|---|---|
| **SS 430** | 13.88 | 13.37 | **10.55** | 9.60 | 9.68 | 9.54 |
| **B540C** | 13.88 | 13.37 | **10.28** | 10.28 | 9.50 | 9.61 |

### 3.2. Change in Time of Corrosion Potential (OCP) during Exposure of Steels to Chloride-Containing Cement Extract Solution

During the change in time of pH cement extract solution (Table 3), an adjustment in the corrosion potential (OCP) values was observed (Table 4). The initial values of SS 430 ($-22.62$ mV) and carbon steel B450C ($-240.48$ mV) indicated that both steels were presenting their passive states at the initial pH of 13.38. However, because of the decrease in pH since the first 24 h, the carbon steel lost its passive state and its potential shifts to more negative values of $\approx -464$ mV at the end of 30 days. In the meantime, the SS 430 shows a tendency to positive values, reaching $\approx 175$ mV at 30 days, manifesting a stable passive state of the surface, attributed mainly to chromium (Cr) as an alloying element introduced to the SS matrix. Reports consider that Cr-reach sites on the surface, which react with air oxygen and are dissolved in the water solution, may create a very thin passive layer of Cr-oxide, of a few atoms, on the SS surface [47–49]. In the absence of chloride ions, our previous study revealed a similar behavior and values of the corrosion potential of SS 430 and carbon steel B450C [34].

**Table 4.** Change in time of the corrosion potential values at open circuit potential (mV vs. SHE) of SS 430 and carbon steel B450C immersed in chloride-containing (5 g L$^{-1}$ NaCl) cement extract solutions for 720 h (30 days).

| Time (h) | Initial | 24 | 168 | 360 | 504 | 720 |
|----------|---------|-----|------|-----|-----|-----|
| SS 430 | −22.62 | −6.91 | 139.97 | 157.31 | 165.50 | 174.96 |
| B450C | −240.48 | −300.61 | −426.38 | −452.28 | −478.50 | −463.81 |

### 3.3. Steel Surface Characterization of Carbon Steel B450C after Exposure to Chloride Containing Cement Extract Solution

Figure 2 shows SEM images of carbon steel B430C surface after exposure for 168 (7 days) and 720 h (30 days) to the cement extract solution (+5 g L$^{-1}$ NaCl), while Table 5 presents the EDS elemental analysis of several zones of interest.

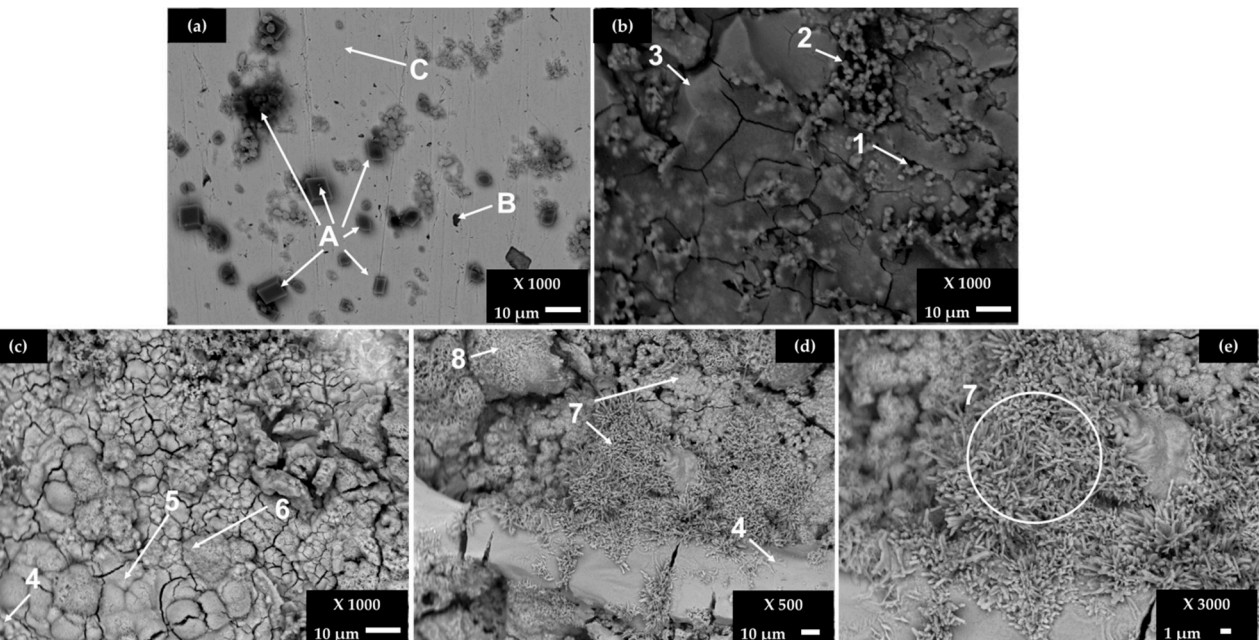

**Figure 2.** SEM images of B450C carbon steel exposed to a cement extract solution (+5 g L$^{-1}$ NaCl): (**a**) a less corroded area and (**b**) a more corroded area after 168 h (7 days); (**c**–**e**) after 720 h (30 days).

**Table 5.** EDS analysis (wt.%) of the B450C carbon steel surface after exposure to the cement extract solution (+5 g L$^{-1}$ NaCl) for 168 h (Figure 2).

| | Element | C | Mn | Si | O | Ca | Cu | Cl | Na | Fe |
|---|---------|-----|-----|-----|------|------|------|------|------|------|
| | **A** | **11.36** | - | - | **50.10** | **36.45** | - | - | 0.35 | 1.73 |
| | **B** | **11.74** | **13.89** | **13.53** | **41.60** | 3.36 | - | - | 0.98 | **14.90** |
| | **C** | 2.73 | 0.80 | 0.34 | 2.02 | - | 1.58 | - | - | 92.53 |
| | 1 | 2.35 | 0.71 | 0.30 | **34.33** | - | 1.13 | - | - | 61.18 |
| | 2 | 3.47 | 1.37 | 0.31 | 37.93 | - | - | - | - | 56.92 |
| **B450C** | 3 | 2.48 | 0.71 | 0.26 | 40.30 | - | - | - | - | 56.25 |
| | 4 | 2.47 | 0.51 | - | 37.12 | - | - | 0.32 | 0.70 | 58.88 |
| | 5 | 2.76 | - | - | 46.47 | - | - | 0.36 | - | 49.49 |
| | 6 | 3.13 | 0.52 | - | 38.19 | - | 1.50 | 0.32 | 0.96 | 55.38 |
| | 7 | 1.86 | - | - | 40.67 | - | - | 4.65 | - | 52.82 |
| | 8 | 3.59 | - | - | 46.78 | - | 1.09 | 2.91 | - | 45.63 |

After the initial period of 7 days (Figure 2a), the surface is not completely covered by a corrosion layer and the EDS analysis reveals the crystals of CaCO$_3$ (labeled as A), as well the particle of high Mn and C content (labeled as B), suggested as Mn3C phase [39,40].

On the area covered by the dense layer of corrosion products (Figure 2b), a variety of iron-oxide morphologies are observed, in the presence of Cu (zones 2 and 3, Table 5). At the end of 30 days (Figure 2c–e) of the experiment, the steel surface is covered completely by a dense layer, presenting micro- and macro-cracks. The iron corrosion products are usually composed by the phases of different oxides and hydroxides ($\alpha$- $\beta$- and $\gamma$-FeOOH), presenting their own chemical and physical properties, as well as a lower density than that of the iron matrix; as they are voluminous, cracks occur. Zones 4 to 8 reveal the high contents of O and Fe, in the presence of chloride, more significantly in zone 7. According to studies, the morphology of iron corrosion products changes in the time, attributed to different phases: sandy crystals, crystalline globules or fine plates look alike to the morphology of $\gamma$-FeOOH (lepidocrocite), while the $Fe_3O_4$ magnetite phase is represented by the rods of different sizes and dark flat regions with circular disks. The $\alpha$-FeOOH goethite grows in globular structure (known as cotton balls) [50–56]. It was also reported that some iron products grow as staggered filaments (filiform corrosion), as observed in zone 7 (Figure 2d,e), extending radially from the chloride-rich zone (more active in corrosion) to the poorer zone [54].

The EDS elemental analysis was collaborated with XRD spectra (Figure 3) analysis, which revealed that the main corrosion products formed on the B450C carbon steel surface (after exposure for 30 days) are $\gamma$-FeOOH (lepidocrocite), $\alpha$-FeOOH (goethite), magnetite ($Fe_3O_4$) and, at lower intensity, crystals of $CaCO_3$ (calcite) appear (Figure 3b).

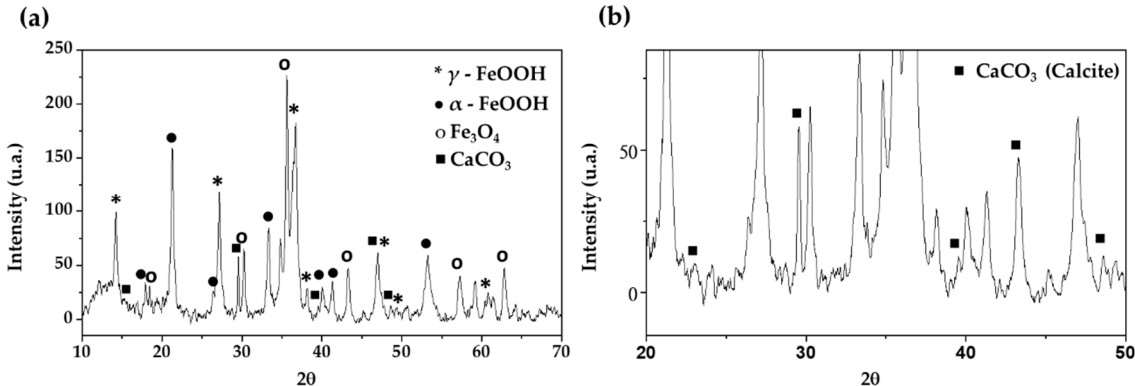

**Figure 3.** XRD spectra of corrosion products formed on B450C carbon steel surface (**a**) and $CaCO_3$ spectra (**b**) after 30 days of exposure to chloride-containing cement extract solution.

They have been many attempts to study the formation mechanism and composition of corrosion products of iron in different environments [50,57–62]. The reported SEM-EDS and X-ray diffraction data indicate that, initially, the nests of $\gamma$-FeOOH (lepidocrocite) are formed, characterized by well defined grains, with membranes that later break, and the secondary phase $\alpha$-FeOOH (goethite) usually grows in the crevices, simultaneously with newly formed $\gamma$-FeOOH grains [60]. Thus, the corrosion product has a multilayer and heterogeneous structure, where crystalline and amorphous films alternate. The $\alpha$-phase has been considered as a transition phase of the Fe-corrosion products, while the $\gamma$-phase is considered as the final by-product of reinforcement [50]. Studies suggest that the transport of oxygen to the iron is impeded in the time, when thicker rust layer is formed, and various ferric oxide/hydroxyde phases may reduce to magnetite ($Fe_3O_4$) [57,58].

### 3.4. Steel Surface Characterization of SS 430 after Exposure to Chloride-Containing Cement Extract Solution

Figure 4 shows SEM images of stainless steel SS 430 surface after exposure for 168 (7 days) and 720 h (30 days) to the cement extract solution (+5 g $L^{-1}$ NaCl), while Table 6 presents the EDS elemental analysis of several zones of interest. Due to the manifested stable passive state on the SS 430 surface (very positive corrosion potential, even the change of pH cement solution, Tables 3 and 4, respectively), a dense layer on the surface was not

observed (compared to that of carbon steel B450C, Figure 2) and only selected areas were covered, mainly by crystals of $CaCO_3$ and NaCl (Table 6).

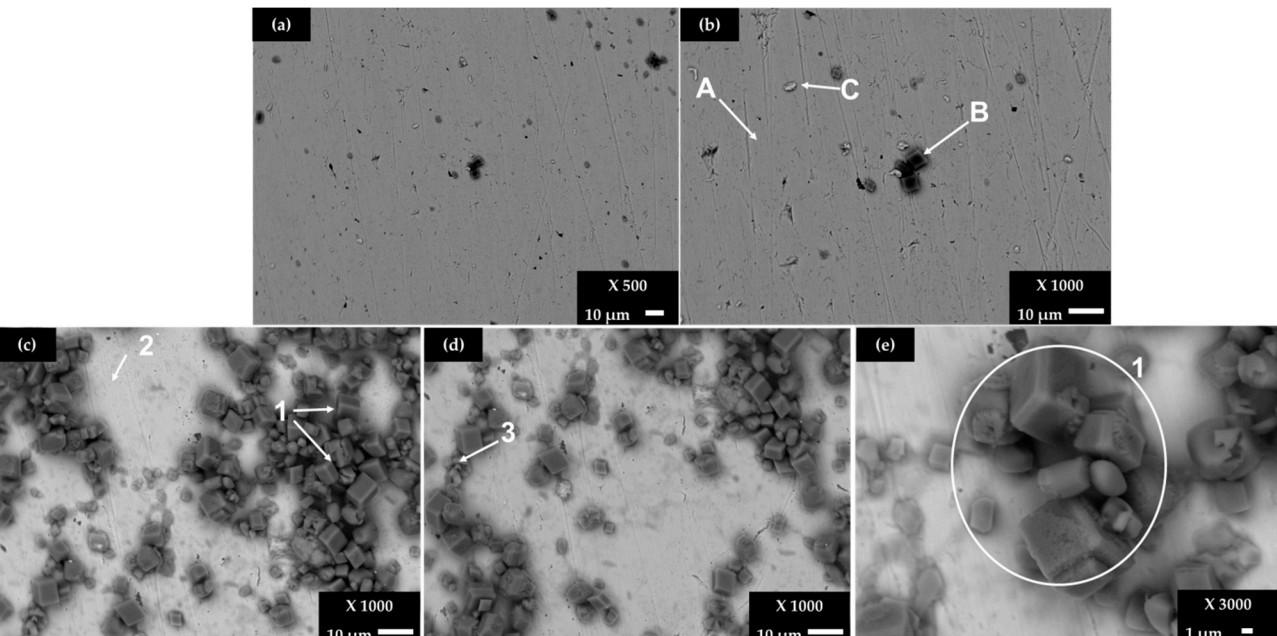

**Figure 4.** SEM images of SS 430 steel exposed to a cement extract solution +5 g $L^{-1}$: for 7 days (**a**) ×500 and (**b**) ×1000; for 30 days (**c**) ×1000, (**d**) ×1000 and (**e**) ×3000.

**Table 6.** EDS analysis (wt.%) of SS 430 carbon steel surface after exposure to the cement extract solution (+5 g $L^{-1}$ NaCl) for 168 h and 720 h (Figure 4).

|  | Element | C | Cr | Si | O | Ca | Cl | Na | Fe |
|---|---|---|---|---|---|---|---|---|---|
|  | A | 2.27 | 6.25 | 0.49 | 1.18 | - | - | - | 79.81 |
|  | **B** | **14.81** | 0.63 | - | **53.66** | 28.78 | - | - | 2.12 |
| **SS 430** | C | 7.32 | 6.76 | 0.38 | 3.63 | - | - | - | 81.90 |
|  | **1** | **15.87** | 1.41 | - | **47.24** | 27.02 | 2.21 | 0.95 | 5.31 |
|  | 2 | 8.08 | 15.92 | 0.30 | 2.80 | 1.03 | 0.25 | - | 71.61 |
|  | **3** | **24.68** | 1.06 | - | 9.99 | 1.87 | **41.10** | **17.64** | 3.65 |

The elemental composition of the A and B zones (Figure 4b) may be attributed to the existence of Fe-Cr-C carbide phases as a part of the steel solid solution [63]. After 7 days of exposure, the crystals of $CaCO_3$ (Figure 4b, zone B) have been deposited on the SS 430 surface, and they continue to be formed at the later period of 30 days (Figure 4c, zone 1), accompanied by the crystals of NaCl, with more significant content in zone 3 (Figure 4c). It seems that the precursor for $CaCO_3$ and NaCl deposition is where the carbon content is higher (local cathodes). On the other hand, first pitting events (localized corrosion) are observed on the SS 403 surface after 30 days of exposure to the chloride-containing cement extract solution.

*3.5. Steel Surface Characterization Damage after Exposure to Chloride-Containing Cement Extract Solution*

After the removal of the corrosion layers, formed after 30 days of exposure of the steels, the SEM images show that the little paths of pitting corrosion (Figure 5a), which are more appreciable in the zoomed area around them (Figure 5b), are visible on the SS 430. The elemental EDS composition is resumed in Table 7. The A zones (Figure 5b) present particles having high contents (wt.%) of Cr (42.36), N (12.61), V (2.94) and C (8.44), which were also present initially on the control sample surface of this steel (Figure 1, zone A, Table 2),

attributed to the Cr-C-N crystal structure and to precipitates of vanadium carbonitrides V (C,N) [38]. The elemental composition of zone B corresponds to the steel matrix. In the meantime, the surface of carbon steel B450C is more damaged (Figure 5b), where zones C and D (Figure 5d) reveal particles with high Mn ($\approx$60–30), Cu ($\approx$16–21) and S contents (wt.%), as elements reported by the supplier and observed on the control sample surface (Figure 1, zone C, Table 2). The deep attack in the steel matrix around zone D (Figure 5d) may be promoted by the presence of the elements Cu and S, which demonstrate activity that is probably cathodic. The E zone shows high content of Fe (78.96) in the presence of C, Mn, O, Cu and S, attributed to the steel matrix.

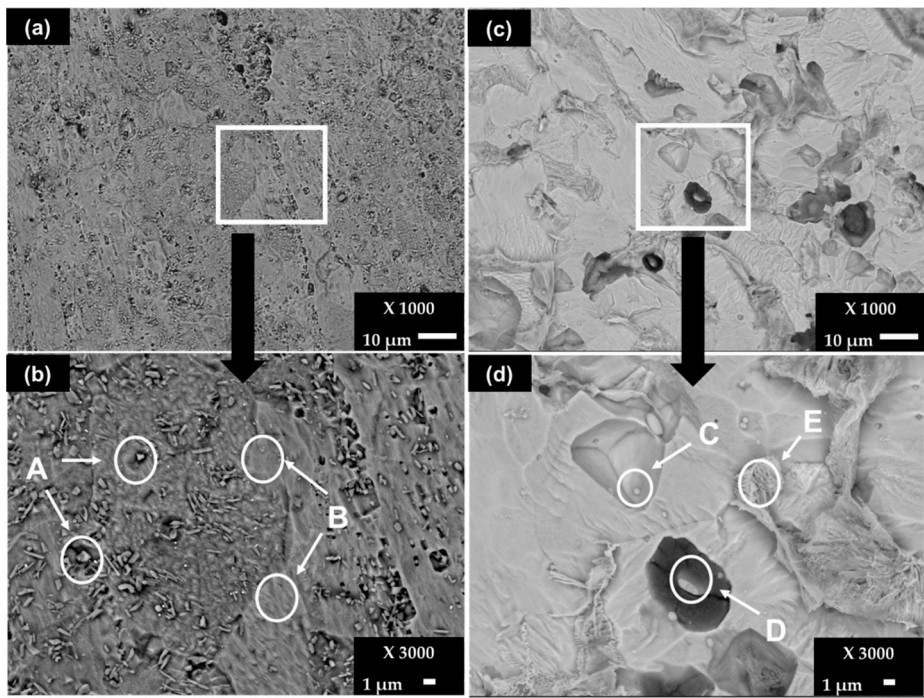

**Figure 5.** SEM images ($\times$1000) of steel surfaces after removal of corrosion layers formed during exposure to chloride-containing cement extract solution for 720 h (30 days): (**a**) SS 430 and (**b**) zoom ($\times$3000); (**c**) B450C and (**d**) zoom ($\times$3000).

**Table 7.** EDS surface analysis (wt.%) of SS 430 and carbon steel B450C after removal of corrosion layers formed during exposure to cement extract solution (+5 g L$^{-1}$ NaCl) for 720 h.

|  | Element | C | Cr | Mn | O | V | Cu | N | S | Fe |
|---|---|---|---|---|---|---|---|---|---|---|
| **SS 430** | A | 8.44 | 42.36 | - | - | 2.94 | - | 12.61 | - | 33.64 |
|  | B | 2.46 | 16.46 | - | - | - | - | - | - | 81.04 |
| **B450C** | C | 4.93 | - | 15.99 | 3.15 | - | 20.75 | - | 13.97 | 41.71 |
|  | D | 10.06 | - | 30.84 | 9.71 | - | 16.28 | - | 21.67 | 11.44 |
|  | E | 12.68 | - | 1.35 | 5.13 | - | 1.34 | - | 0.55 | 78.96 |

### 3.6. X-ray Photoelectron Spectroscopy (XPS) Spectra

In order to correlate the elemental quantification analysis SEM-EDS (Tables 5 and 6) and X-ray diffraction spectra (Figure 3) with the phases present on the studied steel surfaces, XPS was performed. Figure 6 compares the high-resolution spectra of several signals, deconvoluted into chemical states, as the most probable oxidized and non-oxidized components corresponding to layers formed on SS 430 and carbon steel B450C surfaces, after their exposure for 30 days to chloride-containing cement extract solution. The analysis of the spectra was followed according to the procedures suggested by several authors [44,64]. The peak with the highest intensity is O1s, followed by that of Fe2p, Ca2p, C1s, Na1s,

Cr2p and Cl2p. The displayed peak for O1s (Figure 6d) corresponds to $O^{2-}$ in oxides (at 530.2 eV) and $OH^-$ in hydroxides (at 532.1 eV). According to the biding energies [65], they were attributed to $Cr_2O_3$ (at 567.8 eV), formed on the SS 430 surface in contact with atmosphere, and to $Cr(OH)_3$ (at 577.7 eV) corrosion product (Figure 6b), as well to the iron corrosion products of $Fe_3O_4$ (at 709.7 ± 0.1 eV), $\gamma$-FeOOH (at 711.30 eV) and $\alpha$-FeOOH (at 711.79 eV) (Figure 6c), formed on carbon steel B450C surface during the exposure to the chloride-containing cement extract solution. The detection of the Ca2p peak (Figure 6e), together with that of O1s (Figure 6d), can be attributed to the formation of $CaCO_3$ crystals (at 347.06–347.7 eV) on the metal surfaces. The peaks of Cl2p (Figure 6g) and Na1s (Figure 6h) were attributed to NaCl crystals, well observed on the SS 430 surface (Figure 4, Table 5), and also on the carbon B450C surface (Figure 2, Table 5), as suggested by the SEM-EDS analysis. It may be concluded that the XPS registered spectra correlate well with the EDS-SEM analysis corresponding to both steels.

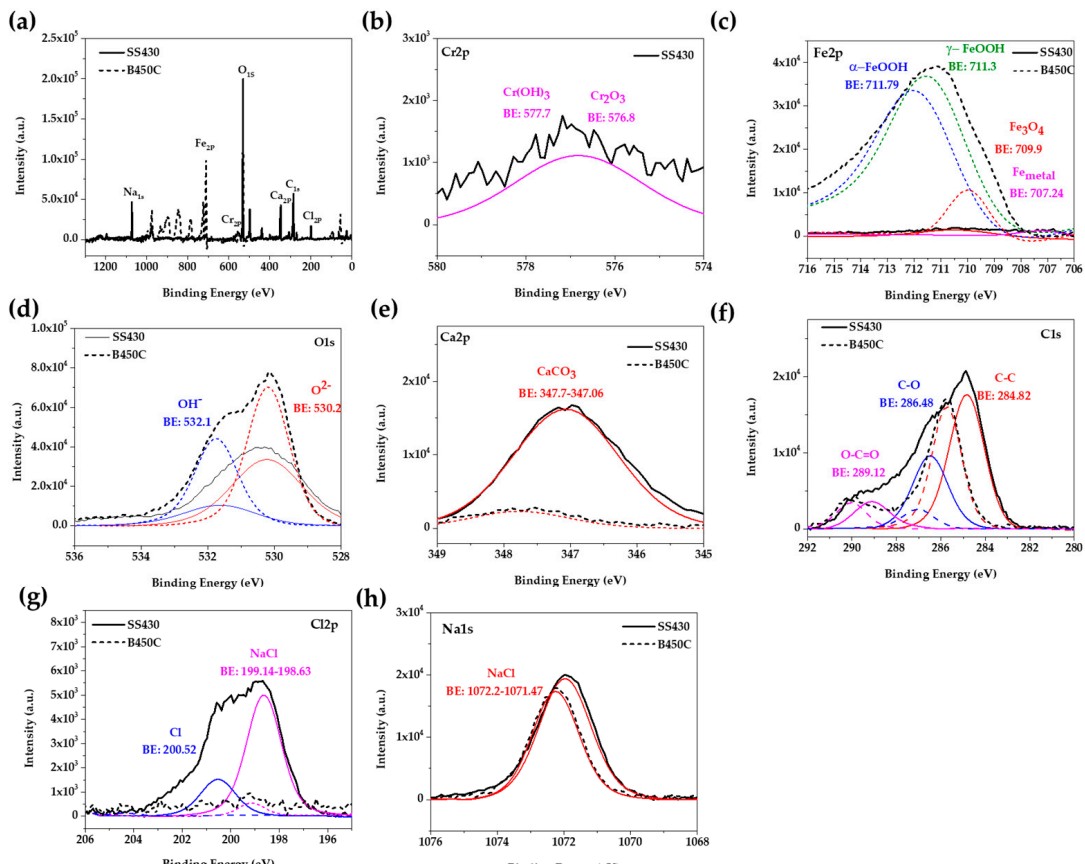

**Figure 6.** Overview of X-ray photoelectron spectroscopy (XPS) spectra acquired from SS 430 and carbon steel B450C after exposure to cement extract solution (+5 g $L^{-1}$ NaCl) for 30 days: (**a**) full spectrum; spectrum for (**b**) Cr2p; (**c**) Fe2p; (**d**) O1s; (**e**) Ca2p; (**f**) C1s; (**g**) Cl2p and (**h**) Na1s.

*3.7. Electrochemical Measurements*

3.7.1. Potentiodynamic Polarization Curves (PDP)

Figure 7 compares the anodic and cathodic potentiodynamic polarization curves (vs. OCP) of SS 430 stainless steel and B450C carbon steel, exposed to the cement extract solution in the presence of chloride ions (5 g $L^{-1}$ NaCl).

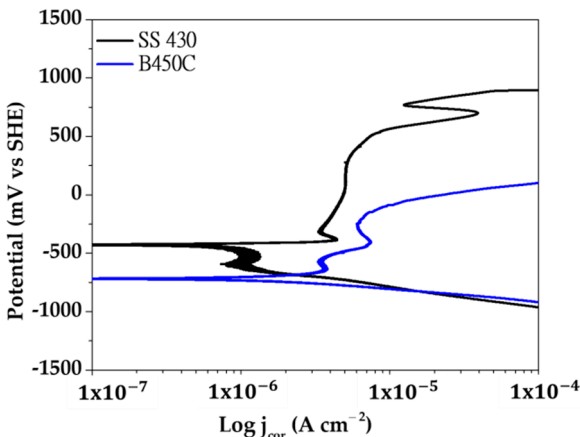

**Figure 7.** Potentiodynamic polarization curves of SS 430 stainless steel and B450C carbon steel exposed to the cement extract solution ($+5.0$ g L$^{-1}$ NaCl).

At a small anodic polarization (Figure 7, black line), the surface of SS 430 enters to a passivation state (passive film formation). Therefore, in a large range of potentials ($-490$ mV up to $+480$ mV) the corrosion current density maintains its low value (1.20 $\mu$A cm$^{-2}$ to 7.07 $\mu$A cm$^{-2}$), followed by an active pitting corrosion stage (increase in the current density). However, upon reaching the anodic potential of $\approx+530$ mV, the surface tries to repassivate and the corrosion rate (the value of the current density) decreases, followed by another return to a corrosion active state (a rapid increases in the corrosion rate). The cathodic process (reduction of $O_2$) in the presence of the native layer of oxides (mainly $Cr_2O_3$) initially presents difficulty, indicated by the tendency of decrease in the value of the current density in the range of potentials from $\approx510$ mV to $-540$ mV. After that period, the cathodic current increases. The processes of passivation, rupture of the passive layer (breakdown) and repassivation of the surfaces of stainless steels have been widely studied, however their mechanisms have not yet been fully understood [66–70]. Studies agree that among the factors that lead to the destruction of the passive layer, there are two that are most important: (1) local acidification in sites (below pH 5), where as a consequence the nucleation of pits may initiate, and (2) defects in the passive layer where aggressive ions, such as chlorides, can penetrate to the Fe-matrix and initiate the corrosion attacks.

In the case of B450C carbon steel (Figure 7, blue line), the active corrosion (oxidation) stage begins immediately with the anodic polarization, and in the range of potentials between $-530$ mV and $-250$ mV the current tries to decrease, due to the formation of the thicker layer of corrosion products. However, at higher anode potentials the corrosion rate (current density) increases significantly. On the other hand, the cathodic current of $O_2$ gas reduction occurs without difficulty, which in fact indicates that multiple cathodic sites (Cu, S, C, in Table 1) are in arrangement on the carbon steel surface.

### 3.7.2. Electrochemical Impedance Spectroscopy (EIS)

Figure 8 compares the Nyquist diagrams for SS 430 and carbon steel B450C, immersed for different periods (up to 30 days) in cement extract solution ($+5$g L$^{-1}$ of NaCl). Since the initial time of exposure the stainless steel (SS 430) tends to semi-linear diffusion impedance (Figure 8a), which is attributed to the diffusion control of the corrosion process because of the passive layer formed on the steel surface [65,71], and a linear slope with an angle of a $\approx45°$ is observed. The passive state of SS 430 is confirmed by the shift of its corrosion potential (OCP) to more and more positive values (Table 4), even when the pH of the chloride-containing cement extract solution (Table 3) reached a lesser alkaline value of $\approx9.6$ (at 30 days). These facts are an indication for the increase in SS 430 corrosion resistance during the 30 days of period of exposure. A similar behavior of SS 430 was observed during the exposure for 30 days to cement extract solution in the absence of chlorides [34].

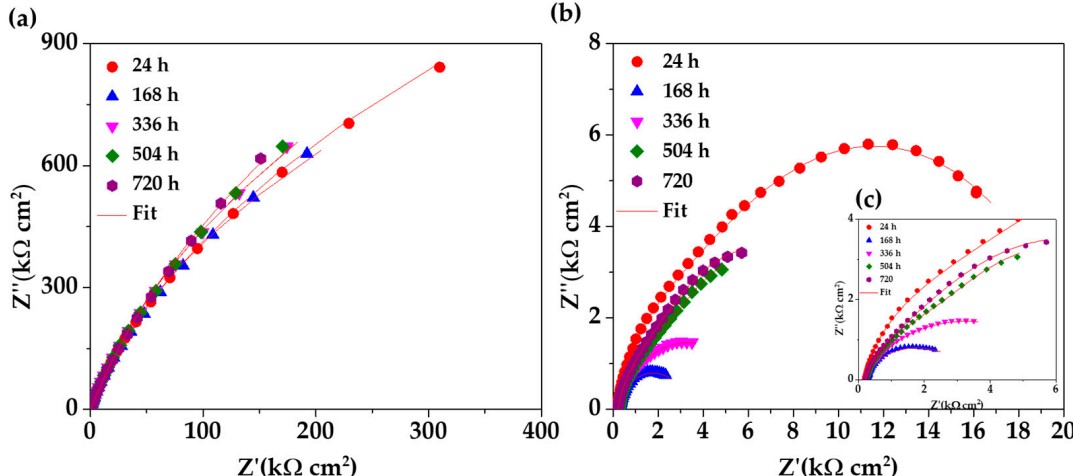

**Figure 8.** EIS Nyquist diagrams with a respective fitting line for SS 430 and carbon steel B450C after different times of immersion in chloride-containing (5 g L$^{-1}$ NaCl) cement extract solution: (**a**) SS 430, (**b**) B450C and (**c**) zoom of B450C (**b**) diagram.

On the other hand, the Nyquist diagrams for the carbon steel B450C (Figure 8b,c) revealed a capacitive behavior and the diameter of the displayed semi-circles decreasing abruptly with time of immersion of the steel in the chloride-containing cement extract solution. This fact indicates that the carbon steel lost its passive state and collaborates well with the shift of the corrosion potential (Table 4) to very negative values ($\approx -464$ mV at the end of 30 days), when the pH of the model solution reached the lesser alkaline value of 9.6 (Table 3).

The significant effect of the chlorides (5 g L$^{-1}$ NaCl), as a part of the cement extract solution) on the corrosion activities of carbon steel B450C, may be appreciated in Figure 9, presenting the Nyquist diagrams in the absence of chlorides [34]. The data reveal that the corrosion resistance of the carbon steel goes down in the presence of chlorides (Figure 8b,c), and, thus, the values of Z″ and Z′ (kΩ·cm$^2$) are more than 10 times lower (Figure 8b,c). The effect of chlorides is observed also in the case of SS 430 activity, even if it is not so remarkable.

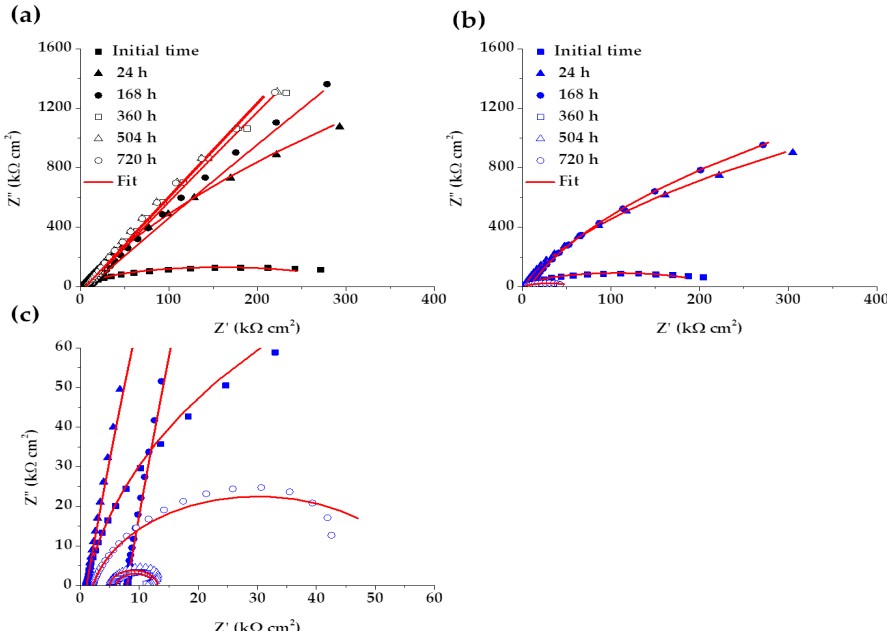

**Figure 9.** EIS Nyquist diagrams with a respective fitting line for SS 430 and carbon steel B450C after different times of immersion in cement extract solution: (**a**) SS 430, (**b**) B450C and (**c**) zoom of B450C (**b**) diagram [34].

These facts confirm the power of the electrochemical impedance technique (EIS), which allows for looking deeper into what happens at the metal-electrolyte interface, whose main characteristics are determined by parameters, such as the resistances of charge transfer and polarization, during the progress of the corrosion process.

The changes in the corrosion activities of the studied steels, presented by the Nyquist diagrams (Figure 8), are also confirmed by the phase angle Bode diagrams (Figure 10). It may be seen that the SS 430 keeps an angle $\approx -85°$ until the end of the experiment (Figure 10a), indicating that the electrode interface is capable of accumulating electrical charges and blocking the migration of aggressive species from the solution (like oxygen and $Cl^-$ ions) through the passive layer, which in fact facilitates the existence of a stable inert thin oxide film on the surface with a low conductivity [71–77]. In the meantime, the lower phase angle values (between $-20°$ and $-40°$) of the carbon steel B450C (Figure 10b) confirmed the capacitive behavior (Figure 8b,c), due to the lost passive state as a consequence of the lesser alkaline pH of the cement extract solution.

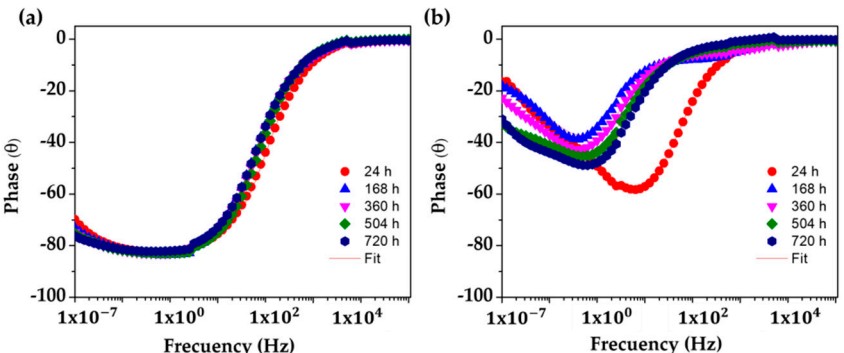

**Figure 10.** EIS Bode diagrams with a phase angle with a respective fitting line for SS 430 and carbon steel B450C after different times of immersion in cement extract solution (+5 g L$^{-1}$ NaCl): (**a**) SS 430, (**b**) B450C.

For a quantitative analysis of EIS results, two equivalent electrical circuit (EC) (Figure 11) have been widely used for the characterization of the electrochemical performance of stainless steels and reinforcing steel exposed to a concrete-simulated pore solution in the literature [78,79]. The first EC (Figure 11a) is a simplified Randles circuit (Figure 11a), describing the electrochemical reactions of metals presenting a passive state (with only one time constant), for example, on a stainless steel surface or on carbon steel in alkaline solutions [80–82]. In this EC, the Rs is the solution resistance at the electrode/electrolyte interface, Rct is the charge transfer resistance and CPE2 is the double-layer capacitance of the interface [83]. The time constant-phase element (CPE) was used instead of the ideal capacitance element, due to the rough and heterogeneous non-ideal surface of the electrode in aggressive environments [80]. Considering the degradation of passive films in the presence of aggressive ions ($Cl^-$) and the possible corrosion attacks in local regions on the carbon steel surface, another EC (Figure 11b) with two time constants (instead of ideal capacitances) was used for fitting the EIS data for the B450C in the presence of chlorides (+5 g/L NaCl) [84–87]. The high-frequency time constant is connected with the defective external oxy-hydroxy Fe(III) layer formed at the interface steel-electrolyte, while the low-frequency time constant corresponds to the charge transfer process that occurred at the interface [72,73]. In this EC (Figure 11b), Rcp represents the resistance of defective passive films and CPE1 correspond to the passive film capacitance [83]. To obtain a better fit, capacitors may be replaced by constant phase elements (CPE), which have an exponential factor $n$, in the range from 0 to 1, where for an ideal capacitor $n = 1$ and $n = 0$ for an ideal resistor [57].

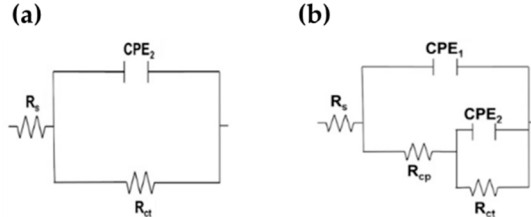

**Figure 11.** Equivalent circuits proposed for SS 430 (**a**) and carbon steel B450C (**b**) exposed to chloride-containing cement extract solution up to 720 h (30 days).

The calculated polarization resistance ($R_p$) of steel [65], which is the sum of $R_{cp}$ and $R_{ct}$ (Equation (1)), may be used as an indicator of the stability of passive films [83]. The change of $R_p$ values over time are presented in Table 8.

$$R_p = R_{cp} + R_{ct} \tag{1}$$

**Table 8.** Fitting parameters obtained from the EIS measurements for SS 430 and carbon steel B450C exposed to chloride-containing cement extract solution up to 720 h (30 days).

| Steel | Time h | $R_{sol}$ $k\Omega cm^2$ | $R_{cp}$ $k\Omega cm^2$ | $CPE_1$ $\mu Ss^n cm^{-2}$ | $n_1$ | $R_{ct}$ $k\Omega cm^2$ | $CPE_2$ $\mu Ss^n cm^{-2}$ | $n_2$ | $R_p$ $k\Omega cm^2$ | $c^2$ $10^{-4}$ |
|---|---|---|---|---|---|---|---|---|---|---|
| | 24 | 0.23 | - | - | - | $5.3 \times 10^3$ | 16.47 | 0.91 | $5.3 \times 10^3$ | 6.04 |
| | 168 | 0.23 | - | - | - | $4.3 \times 10^3$ | 23.09 | 0.92 | $4.3 \times 10^3$ | 10.78 |
| SS 430 | 360 | 0.23 | - | - | - | $5.8 \times 10^3$ | 22.94 | 0.92 | $5.8 \times 10^3$ | 7.20 |
| | 504 | 0.22 | - | - | - | $5.7 \times 10^3$ | 22.73 | 0.92 | $5.7 \times 10^3$ | 2.03 |
| | 720 | 0.24 | - | - | - | $7.3 \times 10^3$ | 23.80 | 0.91 | $7.3 \times 10^3$ | 5.28 |
| | 24 | 0.21 | 6.41 | 56.80 | 0.85 | 20.11 | 191.01 | 0.65 | $2.65 \times 10$ | 1.37 |
| | 168 | 0.23 | 0.29 | 703.25 | 0.46 | 4.24 | 315.88 | 0.96 | 4.53 | 5.21 |
| B450C | 336 | 0.24 | 0.18 | 592.50 | 0.46 | 8.33 | 268.62 | 0.91 | 8.51 | 2.63 |
| | 504 | 0.27 | 6.50 | 625.87 | 0.48 | 7.21 | $2.7 \times 10^3$ | 0.90 | $1.37 \times 10$ | 6.69 |
| | 720 | 0.25 | 4.48 | 511.25 | 0.78 | 9.87 | $1.1 \times 10^3$ | 0.75 | $1.43 \times 10$ | 1.31 |

Due to the stable passive layers formed on the SS 430 (mainly due to $Cr_2O_3$), the charge transfer resistance $R_{ct}$ and polarization resistance $R_p$ of these layers are two-three orders of magnitude higher than those of the cabon steel (Table 8), whose corrosion activity increases after 7 days, when its surface loses the passive state (at the lower pH alkalinity of the solution) and the surface films are less protective in the presence of chlorides.

For the calculation of the passive layer thicknesses (d), the $CPE_2$ values were transformed into the corresponding capacitance values, according to the Brug formula (Equation (2)) [71]. The thickness was calculated from Equation (3) [87], where $\varepsilon_0$ is the vacuum permittivity ($8.85 \times 10^{-14}$ F cm$^{-1}$) and $\varepsilon$ is the dielectric constant of the passive film, which can be assumed as 15.6 for stainless steels [88,89].

$$C = CPE^{\frac{1}{n}} \left( \frac{R_s R_{ct}}{R_s + R_{ct}} \right)^{\frac{1-n}{n}} \tag{2}$$

$$d = \frac{\varepsilon \varepsilon_0 A}{C} \tag{3}$$

Looking for the answer for the effect of chlorides with respect to the electrochemical activities of both steels (the targets of this study), Figure 12 compares the evolution of $R_p$ values and the passive layer thicknesses (*d*), corresponding to the SS 430 and carbon steel B450C, during their immersion in chloride-containing cement extract solution (this study) and in the absence of these aggressive ions [34]. In the absence of chlorides (Figure 12a) the $R_p$ values of SS 430 tend to increase in time; even the pH of the solution decreases in alkalinity [34], as an indication of a stable passive layer on the steel surface. However, in

the chloride-containing cement extract, the polarization resistance ($R_p$) values are relatively constant, being two-three orders of magnitude lower than those in the absence of chloride (Figure 12a).

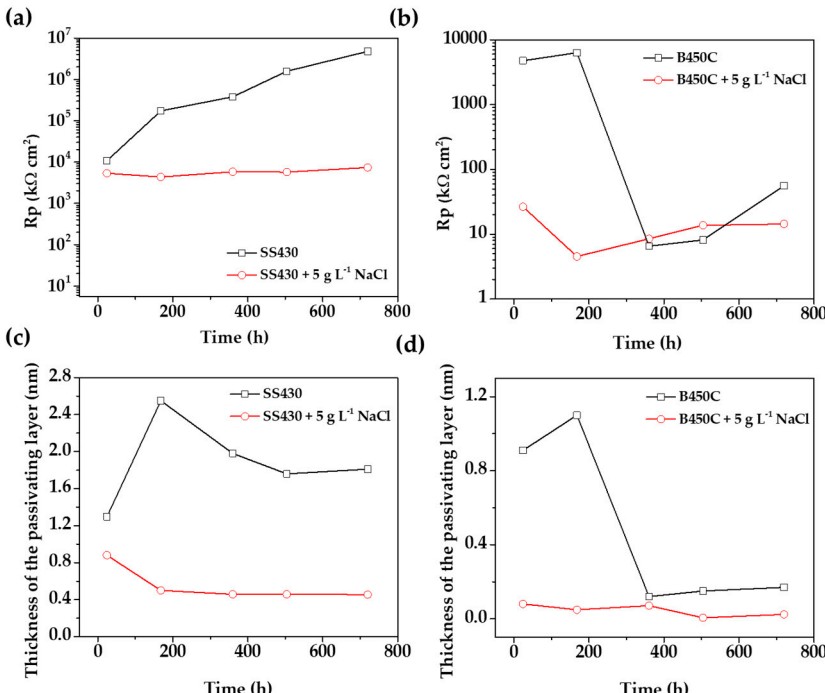

**Figure 12.** (**a**,**b**) Evolution of $R_p$ values and (**c**,**d**) passive layer thickness (*d*) during the immersion of SS 430 and carbon steel B450 to cement extract solution in the presence of chloride-containing cement extract solution (this study) and in the absence of chlorides [34].

In the meantime, in the absence of chlorides, and for the early stages of immersion (up to 7 days) the $R_p$ value of the carbon steel B450C increased (Figure 12b), suggesting the formation of fewer defects/pores and protective films [34]. By prolonging the immersion time (between 15 to 30 days), the $R_p$ decreased very significantly, revealing that the protective thin initial oxide film was lost and replaced by non-protective corrosion product layers. In contrast, in chloride-containing cement extract solution the $R_p$ values were relatively smaller and no significant changes have been observed throughout the immersion test (Figure 12b), revealing that the layers formed on the carbon steel BC450 were non-protective and more susceptible to chloride-localized attacks.

In the absence of chloride ions [34], the thickness of the SS 430 passive layer stabilizes at circa 2 nm and remains around this value until the end of the test (30 days) (Figure 12c). However, in the presence of chlorides (5 g $L^{-1}$ NaCl), the thickness of the passive film is significantly lower and goes down from 0.8 nm to five times lower in thickness, 0.5 nm (Figure 12c). The thinning of the passive layer was possibly caused by the breakdown of the passive layer in the presence of chlorides.

In contrast, in the presence of chlorides the thickness of the passive layer of carbon steel B450C (Figure 12d) tends to disappear after 360 h (30 days), which coincides with the decrease in the pH value of the cement extract solution (Table 3), when the steel lost the passive state. This fact is associated with the localized corrosion on the carbon steel, induced by the chloride aggressive ions. However, in the absence of chlorides [34], the surface was still passivated (up to 168 h) and the greatest thickness was approximately 1.1 nm, decreasing abruptly to circa 0.3 nm (Figure 12d).

## 4. Conclusions

The corrosion activities of commercial low chromium ferritic SS 430 and carbon steel grade B450C were studied during their exposure to chloride-containing (+5 g $L^{-1}$

NaCl) cement extract concrete-pore model solution for 720 h (30 days). SEM-EDS analysis suggested the presence of Cr (C–N) and Si–C phases on SS 430 surface, while on the B450C surface the phases of Si–C and MnS are probably present.

The initial pH of 13.88 tended towards less alkaline values since the first 24 h, maintaining an almost constant pH $\approx$ 9.6 until the end of the test. This decrease in pH was associated mainly with the consumption of $OH^-$ ions, needed for the formation of hydroxides/oxyhydroxides steel corrosion products. Consequently, an adjustment in the corrosion potential (OCP) of carbon steel B450C was observed, shifting to very negative values (loss of the passive state), while the SS 430 OCP showed a tendency to positive values, manifesting a stable passive state of the surface.

At the end of the immersion test (30 days), the B450C surface was completely covered by a dense corrosion layer, presenting micro- and macro-cracks. The EDS analysis suggested Fe-corrosion products presenting a variety of morphology, as well the crystals of $CaCO_3$ and particles of high Mn and C content (Mn-C phases). The EDS elemental analysis was collaborated with XRD spectra and the main corrosion products were considered $\gamma$-FeOOH (lepidocrocite), $\alpha$-FeOOH (goethite), magnetite ($Fe_3O_4$) and, at lower intensity, the crystals of $CaCO_3$ (calcite).

Guveb the manifested stable passive state on the SS 430 surface, a dense layer on the surface was not observed and crystals of $CaCO_3$ and NaCl covered only selected areas. It seems that the precursor for $CaCO_3$ and NaCl deposition is where the carbon content is higher (local cathodes). On the other hand, first pitting events (localized corrosion) are observed on the SS 403 surface after 30 days of exposure.

After the removal of the corrosion layers, formed after 30 days of exposure of the steels, the SEM images showed that the little paths of pitting corrosion are visible on the SS 430. In the meantime, the surface of carbon steel B450C is more damaged and the deep attack was observed around particles of Cu and S content, probably acting as cathodes.

XPS-registered spectra correlate well with the SEM-EDS and XRD analysis, suggesting that after the exposure of the tested steels Fe-oxyhydroxide ($\gamma$-FeOOH and $\alpha$-FeOOH), $Fe_3O_4$ and $Cr(OH)_3$ were formed as corrosion products, as well as $CaCO_3$ and NaCl.

At a small anodic polarization, the surface of SS 430 enters to a passivation state, maintaining a low value (of $\mu A\ cm^{-2}$), followed by an active pitting corrosion stage, although the steel tries to repassivate later. At small cathodic polarization, the reduction process (mainly of $O_2$) presents decrease due to the presence of the native layer of oxides (mainly $Cr_2O_3$). In the case of B450C carbon steel, the active corrosion (oxidation) stage begins immediately with the anodic polarization and later the current tries to decrease, due to the formation of the thicker layer of corrosion products. On the other hand, the cathodic current of $O_2$ gas reduction occurs without difficulty, which in fact indicates that multiple cathodic sites are in arrangement on the surface of carbon steel (Cu, S, C).

For quantitative analysis of EIS (Nyquist and Bode diagrams), two equivalent electrical circuit (EC) were proposed to characterize the corrosion activity of the studied steels and the values of the characteristic parameters at the interface metal-electrolyte. Due to the stable passive layers formed on the SS 430, the charge transfer resistance $R_{ct}$ and polarization resistance $R_p$ are two-three orders of magnitude higher than those of the carbon steel, even at lower alkaline pH and in the presence of chlorides. The calculated polarization resistance ($R_p$) of steel was used as an indicator of the stability of passive films. The $R_p$ values of SS 430 are relatively constant, being two-three orders of magnitude higher than those of the carbon steel B450C.

The calculated thickness (*d*) of the passive layers revealed that SS 430 diminishes from 0.8 nm to $\approx$0.5 nm, with such thinning probably caused by the breakdown of the passive layer in the presence of chlorides. In contrast, the thickness of the passive layer of the carbon steel B450C surface tends to disappear after 30 days, which in fact coincides with the decrease in the pH value of the cement extract solution, followed by the lost state of passivity and localized corrosion induced by the chloride aggressive ions.

**Author Contributions:** Conceptualization and methodology, L.V., D.B. and Á.B. performed the preparation of samples and the corrosion tests; formal analysis of the results, L.V. and D.B.; writing the original draft and editing, L.V., S.F.J. and D.B.; supervision of the project, M.C. and S.L. All correspondence should be addressed to L.V. All authors have read and agreed to the published version of the manuscript.

**Funding:** This research received no external funding.

**Data Availability Statement:** Not applicable.

**Acknowledgments:** David Bonfil acknowledges the Mexican National Council for Science and Technology (CONACYT) for the scholarship granted to him for his M.Sci. study. The authors gratefully thank the National Laboratory of Nano- and Biomaterials (LANNBIO-CINVESTAV) for allowing the use of SEM-EDS and XPS facilities; thanks also go to Victor Rejón Moo, Daniel Aguilar and Wilian Cauich for their support in data acquisition.

**Conflicts of Interest:** The authors declare no conflict of interest.

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
