# Peer review of "Corrosion Activity of Carbon Steel B450C and Low Chromium Ferritic Stainless Steel 430 in Chloride-Containing Cement Extract Solution"

_metals, doi:10.3390/met12010150_

Round 1

Reviewer 1 Report

This is an interesting and well-written paper investigated the corrosion behavior of carbon steel B450C and low chromium ferritic stainless steel 430 in the cement extract solution with chlorides. The following comments should be considered to further improve this paper:

  1. Section 3.6, according to the abovementioned results, active corrosion can be observed for carbon steel B450C after 30 days of exposure to the cement extract solution with chlorides, while stainless steel 430 is still in the passive state. Therefore, what is the reason for comparing the passive film of stainless steel 430 with the corrosion products of carbon steel B450C by XPS measurements?
  2. Why the equivalent electrical circuit with only one time constant was used for fitting the EIS results of stainless steel 430? As confirmed by authors, a passive layer was formed for stainless steel 430 even after 30 days of exposure to low alkaline pore solutions with chlorides, so another time constant of the passive layer should be added in this equivalent electrical circuit. Moreover, Eq.2 should be revised according to the Brug formula.
  3. Fig.12d, I think no passive layer was present for carbon steel B450C after exposure, since a large amount of corrosion products can be observed on the steel surface.

Author Response

Reviewer 1 reports: This is an interesting and well-written paper investigated the corrosion behavior of carbon steel B450C and low chromium ferritic stainless steel 430 in the cement extract solution with chlorides. The following comments should be considered to further improve this paper:

Rev.1: Section 3.6, according to the abovementioned results, active corrosion can be observed for carbon steel B450C after 30 days of exposure to the cement extract solution with chlorides, while stainless steel 430 is still in the passive state. Therefore, what is the reason for comparing the passive film of stainless steel 430 with the corrosion products of carbon steel B450C by XPS measurements?

Veleva L.:  The EDS elemental analysis of the carbon steel (B450C) was collaborated with its XRD spectra (Figure 3) analysis, because the formed corrosion layer after 30 days of exposure was suffucuebt as a quantity. However, in the case of stainless steel (SS 430) it was not like that, so the XRD analysis was not possible. For this reazon, it was necesarry to perform the XPS measurements for both steels, which overview of X-ray photoelectron spectoscopy (XPS) is presented in Figure 6. The XPS registered spectra correlated well with the EDS-SEM analysis corresponding to both steels.

Rev.1: Why the equivalent electrical circuit with only one time constant was used for fitting the EIS results of stainless steel 430? As confirmed by authors, a passive layer was formed for stainless steel 430 even after 30 days of exposure to low alkaline pore solutions with chlorides, so another time constant of the passive layer should be added in this equivalent electrical circuit. Moreover, Eq.2 should be revised according to the Brug formula.

 Veleva L.: The Nyquist diagrams (Fig. 9a) for SS 430 +5 g/l NaCl specimen exhibited only one capacitive loop, suggesting the presence of one single time constant for EIS results. The equivalent circuit fit well to the experimental EIS spectra of SS 430 and it is commonly used to interpret the impedance spectra of stainless steels exposed to concrete simulated pore solution [A-C]. This circuit interprets the passive behaviour of stainless steels in absence or in the presence of chloride alkaline concrete pore, as well as in carbonated Ca(OH)2 [C].

[A] Wang, L.W.; Tian, H.Y.; Gao, H.; Xie, F.Z.; Zhao, K.; Cui, Z.Y. Electrochemical and XPS analytical investigation of the accelerative effect of bicarbonate/carbonate ions on AISI 304 in alkaline environment. Appl. Surf. Sci. 2019, 492, 792–807.

[B] El-Egamy,S.S.; Badaway, W.A. Passivity and passivity breakdown of 304 stainless steel in alkaline sodium sulphate solutions, J. Appl. Electrochem. 2004, 34,1153–1158.

[C] Fajardo, S.; Bastidas, D.M.; Criado, M.; Bastidas, J.M. Electrochemical study on the corrosion behavior of a new low-nickel stainless steel in carbonated alkaline solution in the presence of chlorides. Electrochim. Acta 2014, 129, 160–170.

The used Brug´s formula in our study is correct. Another reference was included in the place of the one cited as [71. Hirschorn, B.; Orazem, M.E.; Tribollet, B.; Vivier, V.; Frateur, A.; Musiani, M. Determination of effective capacitance and film thickness from constant-phase-element parameters Electrochim. Acta. 2010, 55, 6218–6227. https://doi.org/10.1016/j.electacta.2009.10.065].

You can find more information on the interpretation of the equivalent circuit, as well as for the use of Brug´s equation, in the references below:

Xu Jinxia, Wei Jiafeng, Ma Guoxu, Tan Qiping, Effect of MgAl-NO2 LDHs inhibitor on steel corrosion in chloride-free and contaminated simulated carbonated concrete pore solutions, Corrosion Science, Volume 176, November 2020, 108940. https://doi.org/10.1016/j.corsci.2020.108940

Fei Wang, Jinxia Xu, Yi Xu, Linhua Jiang, Guoxu Ma, A comparative investigation on cathodic protections of three sacrificial anodes on chloride-contaminated reinforced concrete, Construction and Building Materials, Volume 246, 20 June 2020, 118476. https://doi.org/10.1016/j.conbuildmat.2020.118476

Joanna Eid, Hisasi Takenouti, Bachir Ait Saadi, Said Taibi, Electrochemical studies of steel rebar corrosion in clay: Application to a raw earth concrete, Corrosion Science, Volume 168, 15 May 2020, 108556. https://doi.org/10.1016/j.corsci.2020.108556

Xu Jinxia, Tan Qiping, Mei Youjing, Corrosion protection of steel by Mg-Al layered double hydroxides in simulated concrete pore solution: Effect of SO42-, Corrosion Science, Volume 163, February 2020, 108223. https://doi.org/10.1016/j.corsci.2019.108223

 Rev.1: Fig.12d, I think no passive layer was present for carbon steel B450C after exposure, since a large amount of corrosion products can be observed on the steel surface.

Veleva L.: We reported that in the presence of chlorides the thickness of the passive layer of carbon steel B450C (Figure 12d) tends to disappear after 360 h (30 days), when the steel lost the passive state. However, in the absence of chlorides the surface was still passivated (up to 168 h) and the greatest thickness was approximately 1.1 nm, decreasing abruptly to circa 0.3 nm (Figure 12d). Such even very low thickness should be considered, suggesting the effect of chlorides.

Reviewer 2 Report

The manuscript entitled „Corrosion Activity of Carbon Steel B450C and Low Chromium Ferritic Stainless Steel 430 in Chloride-Containing Cement Extract Solution” is considered to be relevant to the scope of this journal.

The authors have made a good synthesis of the literature that provides an overview of the research evolution in this area.

However, several points need to be addressed prior to publication of this manuscript. My comments/suggestions are given:

1. The scale must also be added to the enlarged images from Figure 1.

2. More information on the registration of Potentiodynamic polarization curves (PDP) should be specified in the materials and methods section.

3. The studies should be supplemented with electrolyte analysis in which the two alloys were immersed to know exactly their composition and to be able to clearly tell which compounds formed on the surfaces exposed to corrosion.

4. The XRD spectra for CaCO3 must be integrated in Figure 3.

5. XRD spectra of corrosion products formed on stainless steel SS 430 surface must also be presented, after 30 days of exposure to chloride-containing cement extract solution.

6. In Figure 8, the enlarged image should be included in Figure 8b because it is part of it.

7. The fitting lines must have the same color chosen for the experimental data symbols.

8. NaCl is missing from the legend in Figure 10. Must be completed.

9. Is the fitting error 104? I think it is a mistake! See line 469!

10. Conclusions should be written more concisely!

11. The authors' contribution is not written according to the requirements of the journal.

Author Response

Rev.2: Comments and Suggestions for Authors

The manuscript entitled „Corrosion Activity of Carbon Steel B450C and Low Chromium Ferritic Stainless Steel 430 in Chloride-Containing Cement Extract Solution” is considered to be relevant to the scope of this journal. The authors have made a good synthesis of the literature that provides an overview of the research evolution in this area. However, several points need to be addressed prior to publication of this manuscript. My comments/suggestions are given:

Rev.2: The scale must also be added to the enlarged images from Figure 1.

Veleva L.: The scale was also added in Figure 1.

Rev.2 More information on the registration of Potentiodynamic polarization curves (PDP) should be specified in the materials and methods section.

Veleva L.: The information is completed: “Potentiodynamic polarization curves (PDP) were collected after achievement of a stable initial open circuit potential (OCP) of the electrodes immersed in the cement extract solution, by applying polarization to -500 mV and +1500 mV, with a scan rate of 1 mV s-1. The obtained PDP curves were examined with the Gamry Echem Anayst® (version 7.1, Philadelphia, PA, USA).”

Rev.2: The studies should be supplemented with electrolyte analysis in which the two alloys were immersed to know exactly their composition and to be able to clearly tell which compounds formed on the surfaces exposed to corrosion.

Veleva L.: .The required analysis is given in lines 152 – 160.

Rev.2: The XRD spectra for CaCO3 must be integrated in Figure 3.

Veleva L: The spectra for CaCO3 was integrated.

Rev.2: In Figure 8, the enlarged image should be included in Figure 8b because it is part of it.

Veleva L.: The enlarged image was included in Figure 8b, as a part of it.

Rev.2: XRD spectra of corrosion products formed on stainless steel SS 430 surface must also be presented, after 30 days of exposure to chloride-containing cement extract solution.

Veleva L.:  The EDS elemental analysis of the carbon steel (B450C) was collaborated with its XRD spectra (Figure 3) analysis, because the formed corrosion layer after 30 days of exposure was suffucuebt as a quantity. However, in the case of stainless steel (SS 430) it was not like that, so the XRD analysis was not possible. For this reazon, it was necesarry to perform the XPS measurements for both steels, which overview of X-ray photoelectron spectoscopy (XPS) is presented in Figure 6. The XPS registered spectra correlated well with the EDS-SEM analysis corresponding to both steels.

Rev.2: The fitting lines must have the same color chosen for the experimental data symbols.

NaCl is missing from the legend in Figure 10. Must be completed.

Veleva L.: The color of the fitting lines was changed.

Rev.2:  Is the fitting error 104? I think it is a mistake! See line 469!

Veleva L.: You are correct: the negative sign is missing in the text (line 469). However, the values of the fit of c2  present this negative sign (Table 8).

“The values of the characteristic parameters obtained from the EIS measurements are resumed in Table 8 and their fit c2 (10-4) was good in the most cases”.

Rev.2: Conclusions should be written more concisely.

Veleva L.: We would like to keep the conclusions as they are considered.

Rev.2: The authors' contribution is not written according to the requirements of the journal

Veleva L.: “Author Contributions: Conceptualization and methodology, LV;  D.B. performed the preparation of samples and the corrosion tests; formal analysis of the results, L.V. and D.B; writing the original draft and editing, L.V, S. F. and D.B; supervision of the project, M.C. and C.R.  All authors have read and agree to the published version of the manuscript. All correspondence should be addressed to L.V.”

Round 2

Reviewer 1 Report

The commens have been revised by authors. Therefore, it can be accepted in the present form.